# Third-Generation Biomass Crops in the New Era of *De Novo* Domestication

**Christian Wever [1,\*] , David L. Van Tassel [2] and Ralf Pude [1,3]**

[1]   Institute of Crop Science and Resource Conservation (INRES), Renewable Resources, University of Bonn, Klein-Altendorf 2, 53359 Rheinbach, Germany; r.pude@uni-bonn.de

[2]   The Land Institute, 2440 E Water Well Rd, Salina, KS 67401, USA; dlvantassel@gmail.com

[3]   Faculty of Agriculture, Field Lab Campus Klein-Altendorf, University of Bonn, Klein-Altendorf 2, 53359 Rheinbach, Germany

\*   Correspondence: cwever@uni-bonn.de; Tel.: +49-2225-99963-52

**Abstract:** The emerging bioeconomy will increase the need for plant biomass. We call for a third-generation of bioenergy crops, or biomass crops, to help move society towards a sustainable bioeconomy and global food security. Third-generation biomass crops should be capable of producing both food and raw materials. Such flexibility would allow farmers to respond to global markets and buffer global food security. At the same time, third-generation biomass crops need to increase the sustainability of agriculture. To reach such ambitious goals, new biomass crops have to develop *de novo* from promising perennial wild species.

**Keywords:** opinion; *de novo* domestication; biomass crops; perennial bioenergy crops; bioeconomy; third-generation biomass crops

---

The dilemma of devoting productive land to fuel vs. food has made bioenergy crops controversial. The critique of first-generation bioenergy crops is particularly strong [1]. Crops such as maize, rapeseed, sunflower, sugar beet and wheat were domesticated for direct human consumption, or adapted for indirect human nutrition, through the feeding of farm animals. These crops produce large amounts of lipids or carbohydrates that are easily converted to fuels such as biogas, biodiesel or ethanol [2]. However, the production of these annual crops is energy and/or labor intensive. Any energy inputs must be counted against the yield of energy from these crops. It is obvious that the heavy equipment used to till the soil and plant requires fuel and emits $CO_2$, but there are other less obvious energy inputs including the fossil fuel-intensive production of nitrogen fertilizer [3], pumping of water for irrigation and the production of seed for annual replanting. As perennial crops do not require annual replanting and/or soil tillage, the balance between input and harvested energy could be massively improved [4]. Glover et al. (2010), found that a harvested perennial grassland required a 11.75-fold lower energy input than nearby annual cropping systems, with similar output of biomass and protein per hectare [5]. New perennial bioenergy crops were therefore sought.

So-called second-generation bioenergy crops include miscanthus, poplar, alfalfa and switchgrass [6]. Second-generation bioenergy crops represent a shift towards the ideal standard of low-input high-diversity perennial grassland biomass [7]. In addition to the potential for reduced fossil fuel dependence and improved energy return on investment (EROI), perennial crops generate other ecosystem services such as erosion control and resilience during drought [8]. Thanks to their strong, deep, persistent root systems, they may be able to explore a greater soil volume and therefore require fewer inputs of nutrients or irrigation water [9]. Beyond the direct economic and ecological advantages of perennial bioenergy crops, their adoption may defuse ethical concerns associated with bioenergy crops. First, it is possible that these hardy, generally undomesticated, crops can be grown on

marginal or degraded land where food crops are not grown [10]. Second, the fact that these are wild (or nearly wild) species with no history of human use as foods reduces the psychological association of biomass production with global food insecurity. In summary, the advantages of second-generation biomass crops are dominated by the sustainable traits of perennial plants.

The social and ethical controversy about bioenergy crops may not be so easily solved by second-generation bioenergy crops, however. For one thing, there is a second-generation of food crops in development: interest is growing in promoting existing perennial food crops (e.g., fruits, nuts, berries, coconut), perennializing crops such as rice, sorghum and wheat, or domesticating new perennial food crops *de novo* [4,11,12]. *De novo* domestication simply means the domestication of a currently undomesticated plant, in contrast to the ongoing domestication or improvement of existing crops that have been cultivated or harvested since premodern times. Rapid genetic change required for domestication could be achieved using gene editing [13] (as was recently achieved for a wild relative of tomato based on existing knowledge of domestication genes [14]) as well as by phenotypic or genomic selection [15]. Considerable agronomic and cultural/economic adaptations are also required for successful adoption of a new crop [16]. These methods could generate perennial food crops that may also be able to survive on marginal land, although it must be recognized that truly marginal or degraded land will produce low yields of perennial food or energy crops. As the EROI and ecosystem service advantages of second-generation bioenergy and food crops are equivalent, the old food vs. fuel debate may revive, especially as population growth continues to increase demand for both food and fuel while efforts to reduce climate disruption and depletion of natural reserves place growing limits on the availability of fossil hydrocarbons. Renewable substitutes for fuels, lubricants, adhesives, polymers, construction materials, fibers and other raw materials are needed [17]. Plants and waste material flows must form the base of this new, so-called bioeconomy. Cascading use of biomass can also lead to a further improvement of the energy balance [18]. In future, all plant biomass should be used as often as possible as a material, and mainly at the end as an energy source.

We believe that the food-fuel tradeoff cannot be completely evaded. On a planet with finite resources, and especially now with many of these resources depleted or degraded, both biomass and food crop sectors must embrace and grapple with these difficult and unpopular choices. Unfortunately, the bioeconomy cannot sustain the levels of energy consumption those of us in the Global North have become accustomed to in the era of cheap oil. Continued expansion of production of both food and bioenergy will lead to the displacement of wild plant ecosystems, which are important reservoirs of biodiversity and habitat. Fortunately, industrialized nations should be able to dramatically reduce consumption of either food or bioenergy, or both. On the one hand, a high proportion of our croplands are used for livestock production. Dramatic reduction in food waste and meat consumption would free up fertile cropland for production of industrial crops (including bioenergy crops) [17]. We could choose to eat less sugar and, in general, reduce our per-capita caloric intake. On the other hand, improvement in residential architecture and higher population density could increase per capita heating and cooling efficiency. Public transportation, road redesign for the benefit of bicycles and pedestrians, and telecommuting will be needed to substitute for private automobiles, especially in North America where per-capita consumption of motor fuels is currently very high [19].

Here, we call for a third-generation of bioenergy crops, or biomass crops, to help move society towards a sustainable bioeconomy and global food security. We seek to resolve parts of the conflict between biomass production and human nutrition by replacing pure biomass second-generation perennial crops with new, more flexible multipurpose third-generation crops. Like first-generation bioenergy crops, their flexible use for either food or energy may help societies navigate the unpredictable changes in the bioeconomy as some societies incrementally reduce consumption of energy, food, or both. In contrast to the first-generation of biomass plants, however, these plants must be perennials, whose intensive root systems enable them to buffer drought periods and scavenge applied fertilizers more efficiently, making them better choices for achieving maximum sustainability and yield stability in times of climate change. Although third-generation crops could be developed out of orphan crops

or wild relatives of our crops, we see the highest potential for enrichment of agricultural diversity by focusing on genera or families not currently represented in agriculture. This new generation of crops must have the following three characteristics:

1.  An explicitly efficient ideotype. New bioenergy crops must not be allowed to compete with food crops and aquatic ecosystems for irrigation water. Except maybe during initial establishment, third-generation crops must not be irrigated. Nitrogen use efficiency must also be very high, which is the case for the second-generation crop, miscanthus, already [20] Crops should be capable of biological nitrogen fixation, near-100% interception of nitrogen fertilizer or compatible with legume intercropping [21]. Processing of the crop must also be local and energy efficient, a target, which has eluded the production of ethanol from cellulosic crops [22]. Perennial life history is probably the only way to achieve this level of efficiency, but not all perennials are deeply rooted or drought tolerant.

2.  Production of multiple ecosystems services.

    a.  Provisioning services: third-generation biomass crops should be capable of producing both human food (either staple food or high-quality animal feed) and industrial raw materials (fuels, fibers etc.) For example, a perennial crop could be harvested late in the season for edible grain or, another year, the whole biomass could be harvested earlier in the season for biogas production/fodder, or during winter (after grain harvesting) for raw material. Such flexibility would allow farmers to respond to global markets and buffer global food security.

    b.  Regulating services: second-generation bioenergy crops generate regulating ecosystem services (e.g., carbon sequestration [23], reduced soil erosion). Third-generation bioenergy crops must also provide regulating services.

    c.  Supporting services: these include biomass production, but biomass is generally removed in bioenergy crops, making it unavailable for natural food webs. Supporting services were found to be compromised in high productivity stands of the second-generation bioenergy crop miscanthus [19], illustrating the need for third-generation crops. Third-generation crops must generate supporting services beyond primary productivity. For example, they must produce floral resources to support pollinators, overwintering, nesting, or support biodiverse soil food webs [24]. Native crops likely have advantages in supporting local wildlife, however, for exactly the same reason in that they are liable to be more susceptible to specialist pests and pathogens.

3.  Increased landscape biodiversity. Third-generation biomass crops must be different species than the common crops of the target region. In many cases this will require *de novo* domestication of wild plants, substantial improvement in orphan crops or forages that have received little plant breeding attention, or genetic adaptation of crops to new climates (e.g., from temperate to tropical). Diversification of plant species in agricultural landscapes is associated with improved habitat for wildlife [25] and could help make these ecosystems more resilient to climate change and episodic stresses such as droughts or floods. Dramatic species enrichment of global agriculture will require the introduction of previously undomesticated species or genera [26,27]. Our planet's botanical resources are a rich source of novel biochemicals and physiological solutions to serious problems. For example, wild halophytes could help maintain the productivity of agricultural land endangered by salinization from irrigation or by the rising sea level [28,29].

The possibility to adapt these new crops directly during the domestication process to the new needs of the bioeconomy must be seen as a great opportunity. We are still in the beginning of identifying perennial wild species as potential crops. The new perennial cereal *Thinopyrum intermedium* has been used for both grain and biomass, without sacrificing productivity or nutrient availability [30]. *Silphium perfoliatum* L. and *Sida hermaphrodita* Rusby are currently rising

stars among new third-generation bioenergy crops [31–35]. In the case of *Silphium perfoliatum,* we see a multipurpose crop in the future for biomass, edible oil and fodder production, if it will be possible to combine traits of the sister species *Silphium integrifolium* Michx in a hybrid (high biomass, high seed yield, big heads, sustainable production, drought tolerance [34]). *Sida hermaphrodita* can be used for heating, fiber production, biogas and high-quality forage. Status as undomesticated wild plants harbors both challenges and opportunities.

Challenges: a great risk now is the premature transition of new candidates from research to practice. This will negatively affect the cultivation of these potential crops in the long term. Wild accessions of biomass plants are not adapted to the crop conditions in the field. This means that the possibility of farmers having negative practical experiences increases rapidly. If the reputation of the rising stars of the third-generation of biomass crops is ruined, the spread and adoption of *de novo* domesticated material adapted to the arable new crops will be difficult. Other challenges include the low representation of wild perennials in major germplasm banks. This was the case for both *S. perfoliatum* and *S. integrifolium*, and both of the authors have spent considerable efforts to collect, propagate and characterize new germplasm. Fortunately, in these cases, many wild populations have been conserved in situ. However, continued fragmentation of native ecosystems, conversion of wild ecosystems to cropland, pollinator decline, and climate change all threaten the germplasm resources of unexplored new crop candidates.

Opportunities: beginning with wild species we have the opportunity to consciously conserve wild traits that may have been lost during the domestication of today's crops [34]. These traits include associations with mutualistic fungi and microbes, resistance and tolerance to biotic and abiotic stresses, and nutrient/water use efficiency. The gene pools of wild species may include numerous subspecies or ecotypes, and it may be possible to deliberately conserve as much genetic variation as possible during domestication in the genomics era.

In general, a fast *de novo* domestication of the new hopeful crops should be possible [14]. The essential genes for domestication are known in most crop genera and this knowledge could also be used for domestication of the new promising wild species [36–39]. Together with the new omic-possibilities, the hour of plant breeders has come for a sustainable perennial transformation of biomass agriculture.

**Author Contributions:** C.W. and D.L.V.T. mainly contributed to the manuscript as well as writing and original draft preparation; R.P. performed supervision and project administration. All authors have read and agreed to the published version of the manuscript.

**Funding:** This research received no external funding.

**Conflicts of Interest:** The authors declare no conflict of interest.

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
