# Peer review of "Third-Generation Biomass Crops in the New Era of De Novo Domestication"

_agronomy, doi:10.3390/agronomy10091322_

Round 1

Reviewer 1 Report

Abstract: This section needs to be heavily edited for flow and grammar. 

Line 27: remove "mainly long ago". 

Line 30-37: Are there studies that actually quantified this difference between annuals and perennials. It might be good to provide readers with actual numbers so we can gain a sense of a significant quantifiable difference. 

Line 39: Should be rephrased with "the right direction" taken out. 

Line 46: I would change immoral to ethical 

Line 46: Are all second generation crops undomesticated? you previously listed alfalfa and that has undergone domestication. Maybe "primitive" is more appropriate? 

Line 47-49: I'm not sure if i follow this line of reasoning. You are still growing crops that is not for food consumption, so wouldn't the ethical argument still be there. Because you are still using land and resources that can be utilized to help global food insecurities. Additionally, crops like legumes are predominantly grown on marginal lands across the world. 

Line 104-126: Maybe potentially discussing a drawback of third generation crops would be the limited wild germplasm in world gene banks and the limited in situ conservation of wild relatives in nature would lead to a difficulty of identifying third generation crops or transitioning annuals to perennials. 

Maybe adding subsections it the text may help with flow. The transitions between paragraphs was somewhat confusing. 

Author Response

Reviewer 1

Abstract: This section needs to be heavily edited for flow and grammar. 

A: We worked on it and added other aspects, and it should have now a better flow and grammar. Ln15–23

Line 27: remove "mainly long ago". 

A: We removed it. Ln30

Line 30-37: Are there studies that actually quantified this difference between annuals and perennials. It might be good to provide readers with actual numbers so we can gain a sense of a significant quantifiable difference. 

A: We added a new reference . Ln52–55

Line 39: Should be rephrased with "the right direction" taken out. 

A: we changed this. Ln58

Line 46: I would change immoral to ethical 

A: we changed this in Ln65–67

Line 46: Are all second generation crops undomesticated? you previously listed alfalfa and that has undergone domestication. Maybe "primitive" is more appropriate? 

A: We added „generally“ to make clear, that some of the 2nd generation crops could have first steps of domestication. Ln67

Line 47-49: I'm not sure if i follow this line of reasoning. You are still growing crops that is not for food consumption, so wouldn't the ethical argument still be there. Because you are still using land and resources that can be utilized to help global food insecurities. Additionally, crops like legumes are predominantly grown on marginal lands across the world. 

A: You are right, legumes could be grown on marginal lands, but in the end marginal land is limited, too. As well as marginal lands are important for rare natural plant communities. For the third generation of biomass crops like Silphium we have to develop not only Biomass-Silphium, we need a multipurpose crop, so it is time to develop a oilseed crop and biomass Silphium-Hybrid in this case, to become more flexible for human needs.

Line 104-126: Maybe potentially discussing a drawback of third generation crops would be the limited wild germplasm in world gene banks and the limited in situ conservation of wild relatives in nature would lead to a difficulty of identifying third generation crops or transitioning annuals to perennials. 

A: We have added sentences making these points. Thank you for the idea! Ln228–239

Maybe adding subsections it the text may help with flow. The transitions between paragraphs was somewhat confusing. 

A: We changes a little bit and hope it will be now less confusing.

Reviewer 2 Report

Third-Generation Biomass Crops in the New Era of De Novo Domestication

Overall:

  • Very interesting and timely perspective on biomass crops
  • De novo Domestication seems very important to this article, yet you don’t mention it in the abstract, and you wait to introduce the concept until the final three paragraphs
  • There is some language I wouldn’t expect to find in a peer-reviewed piece (“so-called,” “good news,” “immoral”)- I understand this is an opinion piece, but I would recommend seeing if you can elevate the language wherever possible
  • In the abstract, you mention discussing all the ethical problems associated with biomass crops in this paper, but I don’t see any discussion of extensification. Even third generation, dual use biomass crops are likely to produce less edible grain than annual crops would on the same land, and are therefore extensive from a food production perspective. There are rebuttals to this, but I would like to see them fleshed out if your goal is to get into the weeds of the ethics of biomass crop production.
  • You begin framing your paper by discussing the domestication of crops for food and animal feed, however, you do not discuss animal feed as a possible use for biomass crops throughout your paper. Many crops that can be used for bioenergy can be used for animal feed, as many of the characteristics that are desirable for biofuel production are desirable for animal feed. At one point, you even juxtapose bioenergy and livestock production as competing land uses (Ln 71-73). Consider including animal feed as a possible output for bioenergy crops, especially to provide flexibility to producers. One of the challenges with second generation bioenergy crops (cellulosic ethanol) has been that markets have not sustained demand for biomass; however, it is conceivable that livestock ranchers could provide an additional source of demand for biomass.
  • You don’t offer a definition of de novo domestication, which seems important, given its importance to your argument. One definition I have heard used has been the introduction of domestication genes into wild plants. Do you mean to use that specific definition, or do you simply mean any new domestication?

Ln 13- replace “sum” with “sums”

Ln 17- Remove “That is for sure.”

Abstract- Add more detail regarding what a third generation crop is (consider removing “ideally,” in Ln 19, since you are defining third-generation biomass crops herein- it might clear up confusion). Consider introducing de novo domestication.

Ln 27- Were these annual crops originally domesticated for animal consumption, or adapted for animals? Clarify.

Ln 37- Comparing the energy balance of two systems (i.e. stover corn and miscanthus) would really help your case here- this might be hard to find, but many are reluctant to see perennials as productive, and hard facts could go a long way toward convincing them; I’m sure such a juxtaposition would make this piece more citable

Ln 46- “perception that bioenergy crops are immoral”- what do you mean by “immoral?” Also, this seems to come out of nowhere; if you aren’t going to introduce inherit problems with bioenergy crops earlier on I would suggest qualifying this statement somehow or elaborating. I’ve read many bioenergy papers, and I’m familiar with many of the problems associated with bioenergy crops, but I’ve never heard anyone describe them as immoral or anything similar. One simple suggestion could be to rephrase as “their adoption may diffuse concerns associated with bioenergy crops.”

Ln 49- I would suggest adding a concluding sentence to this paragraph

Ln 53/54- these crops seem to fit many of your criteria for third generation biomass crops, especially the vague “domesticating new perennial food crops de novo.” Is there something about crops that are primarily grown for food that qualifies them as second generation? Is it their lack of nutrient/ water efficiency? You may want to set that up, because that would strengthen your argument later on, and it would help the reader understand why you are categorizing all these disparate crops together.

Ln 56- Consider replacing “or” with “and/or”

Ln 60- Remove extra spaces

Ln 63- “In the future”

Ln 64- why not also use biomass as animal feed?

Ln 73- Food waste might be an easier target in terms of reducing calorie consumption- it is very difficult to get individuals to consume fewer calories (see literature in the public health/ medical weight management fields).

Ln 76- don’t forget electric cars!

Ln 84- you repeatedly add miscanthus as an example, and I’m not sure it’s helpful- here, it serves to confuse the reader into thinking perhaps miscanthus is an example of a third-generation biomass crop

Ln 88- Clarify if third-generation biomass crops must have all three of the following services, two of the three, or one of the three.

Ln 89- In line 79, you say third generation biomass crops “must” have these characteristics; here, you say “ideally.” Being able to provide some type of provisioning service seems like the definition of a crop, so consider dropping “ideally.”

Ln 89-94- I would suggest making it necessary for third generation crops to be dual use; most plants create seed or fruit that is useful in some way and biomass that is also useful. I would also suggest adding animal feed to the list of potential uses.

Ln 96- Add closing parentheses.

Ln 98-101- Remove “these… Miscanthus [19].” I understand what you’re saying, but it’s confusing and unnecessary.

Ln 102- Consider adding parenthetical reference to biomass production “primary productivity (biomass production).” This will make up for removing lines 98-101.

Ln 98-103- Consider adding one more example of a supporting service- many biomass crops are grasses, so they will not support pollinators; maybe bird nesting?

Ln 104-111- Did you mean to indent this, as part of the list of requirements for third-generation biomass crops? If so, reconsider. I would suggest that de novo domestication should be a strategy to meet the needs of third-generation biomass crops, but that it should not be a requirement for a crop to be considered a third-generation. Criteria 1 and 2 are clear needs in agriculture, whereas de novo domestication is a strategy. If you wanted to keep three points, you could make criteria 3 “increased landscape biodiversity.” There were only 5,400 acres of miscanthus planted in the entire U.S. in 2017, so, in most regions adding miscanthus to the landscape would add diversity (so far as adding any agronomic monoculture to the landscape can be considered adding diversity). All that to say consider referencing that perennial agriculture in itself is a form of diversity, in that most agriculture is annual. Importantly, consider making the argument that no existing species (or no native species) meets your other criteria, and therefore new species need to be domesticated to meet the need.

Author Response

Reviewer 2

Overall:

  • Very interesting and timely perspective on biomass crops

A: Thank you.

  • De novoDomestication seems very important to this article, yet you don’t mention it in the abstract, and you wait to introduce the concept until the final three paragraphs

A: You are right, we added it to the abstract.

  • There is some language I wouldn’t expect to find in a peer-reviewed piece (“so-called,” “good news,” “immoral”)- I understand this is an opinion piece, but I would recommend seeing if you can elevate the language wherever possible

A: we try to transform the language at some parts, but still try to keep the opinion character…

  • In the abstract, you mention discussing all the ethical problems associated with biomass crops in this paper, but I don’t see any discussion of extensification. Even third generation, dual use biomass crops are likely to produce less edible grain than annual crops would on the same land, and are therefore extensive from a food production perspective. There are rebuttals to this, but I would like to see them fleshed out if your goal is to get into the weeds of the ethics of biomass crop production.

A: We have tried to acknowledge that reduced consumption (or cessation of expanding production) for both food and energy in the Global North. We added a sentence about the negative impact of continued expansion.

  • You begin framing your paper by discussing the domestication of crops for food and animal feed, however, you do not discuss animal feed as a possible use for biomass crops throughout your paper. Many crops that can be used for bioenergy can be used for animal feed, as many of the characteristics that are desirable for biofuel production are desirable for animal feed. At one point, you even juxtapose bioenergy and livestock production as competing land uses (Ln 71-73). Consider including animal feed as a possible output for bioenergy crops, especially to provide flexibility to producers. One of the challenges with second generation bioenergy crops (cellulosic ethanol) has been that markets have not sustained demand for biomass; however, it is conceivable that livestock ranchers could provide an additional source of demand for biomass.

A: we added the fodder topic to this opinion.

  • You don’t offer a definition ofde novo domestication, which seems important, given its importance to your argument. One definition I have heard used has been the introduction of domestication genes into wild plants. Do you mean to use that specific definition, or do you simply mean any new domestication?

A: we added more information Ln78–86

Ln 13- replace “sum” with “sums”

A: we removed this part

Ln 17- Remove “That is for sure.”

A: we remove it Ln18

Abstract- Add more detail regarding what a third generation crop is (consider removing “ideally,” in Ln 19, since you are defining third-generation biomass crops herein- it might clear up confusion). Consider introducing de novo domestication.

A: Thanks for these advise, we done it and it have become clearer now. Ln18, Ln22

Ln 27- Were these annual crops originally domesticated for animal consumption, or adapted for animals? Clarify.

A: You are right. The mentioned crops are adapted/used in a second step for animal feeding. Ln30

Ln 37- Comparing the energy balance of two systems (i.e. stover corn and miscanthus) would really help your case here- this might be hard to find, but many are reluctant to see perennials as productive, and hard facts could go a long way toward convincing them; I’m sure such a juxtaposition would make this piece more citable

A: Please find a comparison at Ln52–55

Ln 46- “perception that bioenergy crops are immoral”- what do you mean by “immoral?” Also, this seems to come out of nowhere; if you aren’t going to introduce inherit problems with bioenergy crops earlier on I would suggest qualifying this statement somehow or elaborating. I’ve read many bioenergy papers, and I’m familiar with many of the problems associated with bioenergy crops, but I’ve never heard anyone describe them as immoral or anything similar. One simple suggestion could be to rephrase as “their adoption may diffuse concerns associated with bioenergy crops.”

A: we changed this in Ln65–67

Ln 49- I would suggest adding a concluding sentence to this paragraph

A: we added one at Ln71–72

Ln 53/54- these crops seem to fit many of your criteria for third generation biomass crops, especially the vague “domesticating new perennial food crops de novo.” Is there something about crops that are primarily grown for food that qualifies them as second generation? Is it their lack of nutrient/ water efficiency? You may want to set that up, because that would strengthen your argument later on, and it would help the reader understand why you are categorizing all these disparate crops together.

A: We define it a bit more in Ln130–150, we hope it became clearer now.

Ln 56- Consider replacing “or” with “and/or”

A: replaced Ln39

Ln 60- Remove extra spaces

A: removed Ln60

Ln 63- “In the future”

A: removed Ln91

Ln 64- why not also use biomass as animal feed?

A: we added this thought in Ln216,164,198, 220

Ln 73- Food waste might be an easier target in terms of reducing calorie consumption- it is very difficult to get individuals to consume fewer calories (see literature in the public health/ medical weight management fields).

A: Someday there won’t be a choice. And we are talking about ethics, not “easy choices.” But food waste is a good idea. Ln121

Ln 76- don’t forget electric cars!

A: For us it is not clear, how sustainable electric cars are. There is still the need of rare-earths. Maybe it will become better, if we are able to recycle them. So we dislike to add them in the opinion. We hope this is fine.

Ln 84- you repeatedly add miscanthus as an example, and I’m not sure it’s helpful- here, it serves to confuse the reader into thinking perhaps miscanthus is an example of a third-generation biomass crop

A: We done this, because there is much knowledge about miscanthus. But now we make it clearer in Ln139

Ln 88- Clarify if third-generation biomass crops must have all three of the following services, two of the three, or one of the three.

A: we clarify it in Ln145

Ln 89- In line 79, you say third generation biomass crops “must” have these characteristics; here, you say “ideally.” Being able to provide some type of provisioning service seems like the definition of a crop, so consider dropping “ideally.”

A: we deleted Ln146

Ln 89-94- I would suggest making it necessary for third generation crops to be dual use; most plants create seed or fruit that is useful in some way and biomass that is also useful. I would also suggest adding animal feed to the list of potential uses.

A: we added this thought in Ln147,150, 176

Ln 96- Add closing parentheses.

A: we added in Ln154

Ln 98-101- Remove “these… Miscanthus [19].” I understand what you’re saying, but it’s confusing and unnecessary.

A: we hope it become better know in Ln159

Ln 102- Consider adding parenthetical reference to biomass production “primary productivity (biomass production).” This will make up for removing lines 98-101.

A: we hope it become better know in Ln159

Ln 98-103- Consider adding one more example of a supporting service- many biomass crops are grasses, so they will not support pollinators; maybe bird nesting?

A: in the third generation we see less grasses, but we added overwintering and nesting as important points

Ln 104-111- Did you mean to indent this, as part of the list of requirements for third-generation biomass crops? If so, reconsider. I would suggest that de novo domestication should be a strategy to meet the needs of third-generation biomass crops, but that it should not be a requirement for a crop to be considered a third-generation. Criteria 1 and 2 are clear needs in agriculture, whereas de novo domestication is a strategy. If you wanted to keep three points, you could make criteria 3 “increased landscape biodiversity.” There were only 5,400 acres of miscanthus planted in the entire U.S. in 2017, so, in most regions adding miscanthus to the landscape would add diversity (so far as adding any agronomic monoculture to the landscape can be considered adding diversity). All that to say consider referencing that perennial agriculture in itself is a form of diversity, in that most agriculture is annual. Importantly, consider making the argument that no existing species (or no native species) meets your other criteria, and therefore new species need to be domesticated to meet the need

A: Thank you for this suggestion. We are trying to incentivize domestication here rather than simply using gene editing to make perennial corn or sorghum. We have enough corn and sorghum and yes, we should substitute the existing corn and sorghum with perennial corn and sorghum, but it would be even better to have perennial corn (for traditional food) and some completely different crop as a bioenergy crop.  Or in a place where alfalfa is common, it would not be “diversification” to find a way to turn alfalfa into a human food, although there might be other good reasons to do that. We attempted to soften this requirement. Do you think it works?

Reviewer 3 Report

An article deals with domestication of new biomass crops. New perspectives on the use of biomass are much needed. The current situation is unsustainable. The scope of the manuscript is quite small nevertheless brings interesting information and opinions.

Please see my comments and recommendations:

I would prefer affiliation to be written in English.

Line 100: Supporting services trade-off with biomass yield in the second-generation crop Miscanthus – unclear statement

Some examples of EROI for first and second generation bioenergy crops would better draw the needs for alternatives.

Surely more research should be performed, however I would expect more species you recommend as a potential third generation bioenergy crops.

Silphium perfoliatum L is pointed as a possible alternative. However, we should consider its potential invasiveness.

After corrections I agree to publish this manuscript.

Author Response

Reviewer 3

An article deals with domestication of new biomass crops. New perspectives on the use of biomass are much needed. The current situation is unsustainable. The scope of the manuscript is quite small nevertheless brings interesting information and opinions.

Please see my comments and recommendations:

I would prefer affiliation to be written in English.

A: we change it in Ln5–10

Line 100: Supporting services trade-off with biomass yield in the second-generation crop Miscanthus – unclear statement

A: We reworded this sentence. Ln173–175

Some examples of EROI for first and second generation bioenergy crops would better draw the needs for alternatives.

Surely more research should be performed, however I would expect more species you recommend as a potential third generation bioenergy crops.

A: We added Kernza as an example. The point is that researchers need to re-examine many wild species. Ln207–210

Silphium perfoliatum L is pointed as a possible alternative. However, we should consider its potential invasiveness.

A: Of course every new plant species have the potential to be invasive. But there is not any publication demonstrating it for Silphium perfoliatum L. Due to the fact this species is grown in Europe for over 250 years (Antoine Gouan mentioned it cultivating 1762 in hortus regius monspeliensis), we did not expect any invasiveness now.

After corrections I agree to publish this manuscript.

Reviewer 4 Report

General remarks:

In this opinion paper, the authors call for a “third generation” of bioenergy/biomass crops. The contribution suggests essential characteristics that should guide the development of future biomass production systems. I particularly liked the idea to elaborate the characteristics of future bioenergy/biomass crops on the basis of the well-established classification of first- and second-generation crops. However, I feel that more detail is required in the manuscript to justify the call for a new class of third-generation crops.

In my opinion, more detail is required on two aspects: First, it should be made clearer, why a third generation is required at all. Which kind of issues of second-generation crops can and will be solved by third-generation crops? Second, the differences between second- and third-generation have to be made clearer. So far, the differentiation is rather vague and the “second-generation crop Miscanthus” could also be considered a third-generation crop. This could be refined by providing additional detail on the suggested characteristics and by providing examples which refer to “the rising stars” Silphium perfoliatum L. and Sida hermaphrodita Rusby. In addition to these rather general aspects, the authors will find some more specific remarks in the following:

Specific remarks:

Abstract

Line 17: “That is for sure” – Is this really necessary? It sounds rather unscientific.

Main text

Line 50-54: It is difficult to grasp the idea of the two sentences. Perhaps,  these sentences could be rephrased to the readability.

Line 54: Most readers won’t be familiar with de novo food crops. The concept should be shortly introduced and some examples would benefit the reader.

Line 78: Following the previous paragraph, it is not completely clear, why it is necessary to call for a third generation of bioenergy crops? The argument that they will “help to move society towards a sustainable bioeconomy and global food securty” sounds rather vague. How can third-generation crops actually contribute to these goals?

Line 89-93: Please clarify: Is mutlifunctionality a defining characteristic of third-generation crops? How do Silphium perfoliatum L. and Sida hermaphrodita Rusby fit to this criterium? I agree that both crops could be used for bioenergy and production and to a limited extent also for feed. However, I don’t see how they could be used to buffer global food security.

Line 104-111: As previously mentioned, “de novo domestication” should be introduced/defined somewhere in the manuscript: What is the exact meaning of de novo here? Which kind of plants would be (not) included? Are miscanthus and pawlonia examples? Are only species considered which are so presently undomesticated? The authors could provide some more detail here.

Besides these aspects, I wonder if priority should be given to regionally-dedicated de novo domestication. Obviously, the increase of crop diversity in agricultural landscapes improves the general requirements for biodiversity.

However, it should be kept in mind, that, e.g., Silphium perfoliatum L. which originates in North America, is mainly visited by pollinators such as honey bees, while more specialised polinators might not benefit from it (Dauber,  J.,  A.L.  Müller,  S.  Schittenhelm,  B.  Schoo,  Q.  Schorpp,S. Schrader,  S.  Schroetter,  2016:  Agrarökologische  Bewertungder  Durchwachsenen  Silphie  (Silphium  perfoliatum  L.)  als  eineBiomassepflanze  der  Zukunft.  Schlussbericht  (FKZ  22004411).Braunschweig:  Johann  Heinrich  von  Thünen  Institut.  http://www.fnr.de/index.php?id=11151&fkz=22004411    (Stand:    30.August 2016). A de novo crop, native in Europe, could be more beneficial in this regard.

Line. 113-114: As stated before, a clearer definition of third-generation crops would benefit the differentiation of second- and generation crops.

Author Response

Reviewer 4

General remarks:

In this opinion paper, the authors call for a “third generation” of bioenergy/biomass crops. The contribution suggests essential characteristics that should guide the development of future biomass production systems. I particularly liked the idea to elaborate the characteristics of future bioenergy/biomass crops on the basis of the well-established classification of first- and second-generation crops. However, I feel that more detail is required in the manuscript to justify the call for a new class of third-generation crops.

In my opinion, more detail is required on two aspects: First, it should be made clearer, why a third generation is required at all. Which kind of issues of second-generation crops can and will be solved by third-generation crops? Second, the differences between second- and third-generation have to be made clearer. So far, the differentiation is rather vague and the “second-generation crop Miscanthus” could also be considered a third-generation crop. This could be refined by providing additional detail on the suggested characteristics and by providing examples which refer to “the rising stars” Silphium perfoliatum L. and Sida hermaphrodita Rusby. In addition to these rather general aspects, the authors will find some more specific remarks in the following:

Specific remarks:

Abstract

Line 17: “That is for sure” – Is this really necessary? It sounds rather unscientific.

A: you are right, but it is an opinion, but we delete it for a more scientific language Ln18

Main text

Line 50-54: It is difficult to grasp the idea of the two sentences. Perhaps,  these sentences could be rephrased to the readability.

A: We changed it.

Line 54: Most readers won’t be familiar with de novo food crops. The concept should be shortly introduced and some examples would benefit the reader.

A: We added a short example/definition at Ln78–86

Line 78: Following the previous paragraph, it is not completely clear, why it is necessary to call for a third generation of bioenergy crops? The argument that they will “help to move society towards a sustainable bioeconomy and global food securty” sounds rather vague. How can third-generation crops actually contribute to these goals?

A: We added some sentences to make it clearer in Ln131–142

Line 89-93: Please clarify: Is mutlifunctionality a defining characteristic of third-generation crops? How do Silphium perfoliatum L. and Sida hermaphrodita Rusby fit to this criterium? I agree that both crops could be used for bioenergy and production and to a limited extent also for feed. However, I don’t see how they could be used to buffer global food security.

A: You are right, Sida hermaphrodita has only one second use as fodder, but although to fiber production (and heating). Silphium perfoliatum is much more usable. Additionally to its use a fodder, we see the possibility to create a multipurpose biomass/oilseed, fodder crop out of it. We added some sentences Ln210–218

Line 104-111: As previously mentioned, “de novo domestication” should be introduced/defined somewhere in the manuscript: What is the exact meaning of de novo here? Which kind of plants would be (not) included? Are miscanthus and pawlonia examples? Are only species considered which are so presently undomesticated? The authors could provide some more detail here.

A: We added a short example/definition of de novo domestication at Ln78–86. For the authors miscanthus belongs mainly to the second generation, because it could not used for any other purpose than biomass/energy. Paulownia could be belong to the third generation as their leaves could be used as fodder, but it depends on the way of cultivation, if it will come to flower we see much more benefits as in cropping plantations. We added a sentence to define the origin of de novo domesticated biomass crops of the third generation. Ln141–144

Besides these aspects, I wonder if priority should be given to regionally-dedicated de novodomestication. Obviously, the increase of crop diversity in agricultural landscapes improves the general requirements for biodiversity.

However, it should be kept in mind, that, e.g., Silphium perfoliatum L. which originates in North America, is mainly visited by pollinators such as honey bees, while more specialised polinators might not benefit from it (Dauber,  J.,  A.L.  Müller,  S.  Schittenhelm,  B.  Schoo,  Q.  Schorpp,S. Schrader,  S.  Schroetter,  2016:  Agrarökologische  Bewertungder  Durchwachsenen  Silphie  (Silphium  perfoliatum  L.)  als  eineBiomassepflanze  der  Zukunft.  Schlussbericht  (FKZ  22004411).Braunschweig:  Johann  Heinrich  von  Thünen  Institut.  http://www.fnr.de/index.php?id=11151&fkz=22004411    (Stand:    30.August 2016). A de novo crop, native in Europe, could be more beneficial in this regard.

A: You are right, but in this case Silphium was just used as an example. Of course, we need to identify much more (native) species for biodiversity with different flower shapes and flowering times. We add a hint to this problematic at Ln141–144,207–208

Line. 113-114: As stated before, a clearer definition of third-generation crops would benefit the differentiation of second- and generation crops.

A: We added some sentences to make the definition clearer in Ln131–144

Round 2

Reviewer 4 Report

The authors have thoughtfully replied to the comments made.